# Is Fibersol-2 efficacious in reducing duration of watery diarrhea and stool output in children 1–3 years old? A randomized, parallel, double-blinded, placebo-controlled, two arm clinical trial

Abu Sadat Mohammad Sayeem Bin Shahid[1]*, Shahnawaz Ahmed[2], Sampa Dash[3], Yuka Kishimoto[4], Sumiko Kanahori[4], Tahmeed Ahmed[1], Abu Syed Golam Faruque[1], Mohammod Jobayer Chisti[1]

1 Nutrition and Clinical Services Division, International Centre for Diarrhoeal Disease Research, Dhaka, Bangladesh, 2 The University of Queensland, Brisbane, QLD, Australia, 3 Child Health Research Foundation, Mirzapur, Bangladesh, 4 Matsutani Chemical Industry Co. Ltd, Itami City, Hyogo, Japan

* sayeem@icddrb.org

**Data Availability Statement:** The data set contained personal information of the study

## Abstract

### Background

Fibersol-2 has innumerable beneficial effects on human health. It is a fermentable, non-viscous, water-soluble, indigestible dextrin containing 90% dietary fiber produced from corn starch. We aimed to evaluate whether additional intake of Fibersol-2 along with oral rehydration solution treatment can reduce the duration of watery diarrhea and daily stool output in children 1–3 years as well as recovery of such children within 72 hours, compared to placebo.

### Methods

This placebo-controlled double-blinded, randomized parallel two arm trial conducted in Kumudini Women's Medical College Hospital in rural Bangladesh between March and October, 2018 used 5 gm of either Fibersol-2 or placebo dissolved in 50-ml drinking water which was given orally to ninety-two children with watery diarrhea on enrollment twice daily for a period of 7 days. Randomization was done using a randomization table. We randomly allocated 45 (49%) and 47 (51%) children in Fibersol-2 and placebo groups, respectively. Outcome measures were duration of resolution of watery diarrhea, daily stool output and the proportion of children recovered within 72 hours. Primary and safety analyses were by intention to treat. This trial was registered at ClinicalTrials.gov, number NCT03565393.

### Results

There was no significant difference observed in terms of duration of resolution of diarrhea (adjusted mean difference 8.20, 95% CI -2.74 to 19.15, p = 0.14, adjusted effect size 0.03); the daily stool output (adjusted mean difference 73.57, 95% CI -94.17 to 241.32, p = 0.38,

participants. Our institutional review board will not have the provision to disclose any kind of information. Thus, our policy is not to make availability of the data set in the manuscript, the supplemental files, or a public repository. However, data related to this manuscript are available upon request and for researchers who meet the criteria for access to confidential data may contact with Ms. Armana Ahmed (armana@icddrb.org) to the research administration of icddr,b (http://www.icddrb.org/).

**Funding:** This research study was funded by Matsutani Chemical Industry Company Limited, Japan on behalf of ADM/Matsutani LLC, USA in the form of grants. The International Centre for Diarrhoeal Disease Research, Bangladesh receives unrestricted support from the Government of the People's Republic of Bangladesh, Global Affairs Canada, the Swedish International Development Cooperation Agency and the UK Department for International Development. The funders had no role in study design, data collection and analysis, decision to publish, or preparation of the manuscript.

**Competing interests:** Yuka Kishimoto and Sumiko Kanahori are employed by Matsutani Chemical Industry Co Ltd, Hyogo, Japan. There are no patents, products in development or marketed products associated with this research to declare. This does not alter our adherence to PLOS ONE policies on sharing data and materials.

**Abbreviations:** WHO, World Health Organization; PHGG, Partially hydrolized guar gum; ORS, Oral rehydration solution; MUAC, Mid upper arm circumference.

adjusted effect size 0.33) and the proportion of children recovered within 72 hours (adjusted odds ratio 0.49, 95% CI = 0.12 to 1.96, p = 0.31, adjusted risk difference -0.06 (95% CI -0.19 to -0.06), after regression analysis between Fibersol-2 and placebo.

## Conclusion

No beneficial role of Fibersol-2 was observed in diarrheal children aged 1–3 years.

## Trial registration

This trial is registered at ClinicalTrials.gov, number NCT03565393. The authors confirmed that all ongoing and related trials for this drug/intervention are registered. https://clinicaltrials.gov/ct2/show/NCT03565393.

## Introduction

Dietary fiber, a non-digestible carbohydrate, has been used in decades for the beneficial effect on health with physiological importance because such compounds have low energy values. These indigestible carbohydrates generally reach the large intestine in an undigested and unabsorbed forms and are often used in many low-calorie food and beverages [1–3].

Researchers have stated that dietary fiber, especially digestive-resistant maltodextrin has innumerable beneficial effects on human health, such as improving intestinal regularity by increasing faecal bulk, stimulating peristalsis and shortening gastrointestinal transit time [4,5]. Fibersol-2 (resistant maltodextrin or indigestible dextrin) is a non-viscous, water-soluble, fermentable dietary fiber produced from corn starch.

A prebiotic beneficially affects the host by enhancing the growth of large bowel bacteria, including *Bifidobacterium* and *Lactobacillus*, thus conferring beneficial effects on the health of the host [6]. Prebiotics are known to result in a decrease in pathogenic bacteria, such as *Clostridium perfringens*. They can act by decreasing pH causing increased production of short chain fatty acid (SCFA) resulting in enhanced competition for nutrients. Fibersol-2 reaches the large intestine and half of it undergoes fermentation by intestinal flora [7,8]. Ohkuma *et al*. in 1990 observed changing pattern of microbial flora caused by administration of resistant maltodextrin. and further studies conducted by other researchers confirmed that Fibersol-2 has pre-biotic activity in humans [9–11]. Hypothetically, Fibersol-2 as a prebiotic is assumed to exert the beneficial effect on mucosal immune response to the intestine. The evidence showed that partially hydrolyzed guar gum (PHGG) with added oral rehydration solution (ORS) can enhance early recovery of acute diarrhea in severely malnourished children in terms of reducing the duration of diarrhea and stool output [12].

Although fiber rich diet is widely recommended, the efficacy of fiber supplements has not been tested sufficiently in children. Fibers are expected to be fermented by colonic bacteria producing SCFAs those will stimulate sodium and water absorption in the colon leading to early recovery from diarrhea [13].

According to a systematic review, children admitted to hospital with diarrhea and dehydration, reduced osmolarity rehydration solution with diarrhea is associated with reduced need for unscheduled intravenous infusions, lower stool volume, and less vomiting in comparison with standard world health organization (WHO) rehydration solution. Children with acute diarrhea, therefore, may be benefited from a reduced osmolarity ORS [14–17]. Dietary fibers

can act by changing the nature of the contents of the gastrointestinal tract and by changing how other nutrients and chemicals are absorbed [18]. Advantages of consuming fiber are the production of healthy compounds during fermentation, increased bulk of stool, softening of stool, shortening of transit time through the intestinal tract, blocking of intestinal mucosal adherence, translocation of potentially pathogenic bacteria, and modulation of intestinal inflammation [19–23].

We aimed to examine whether additional intake of Fibersol-2 along with ORS can reduce the duration of watery diarrhea and daily stool output in children aged 1–3 years as well as recovery of such children within 72 hours, compared to placebo.

## Materials and methods

### Ethical consideration

Institutional review board of International Centre for Diarrhoeal Disease Research, Bangladesh approved the study (PR-16091). Informed written consent was obtained from parents or caregivers prior enrolling the participating children.

### Study design

A placebo-controlled, randomized, double-blinded parallel clinical trial to examine whether Fibersol-2 along with ORS can reduce the duration of watery diarrhea and daily stool output in children aged 1–3 years as well as proportion of such children recovered within 72 hours was conducted during the period of March to October, 2018. After screening, ninety-five children with acute watery diarrhea were randomly assigned to either groups, irrespective of sex and socio-demographic background (Fig 1).

### Eligibility criteria for the clinical efficacy trial

#### Inclusion criteria.

i.  Children aged 1–3 years of either sex having acute watery diarrhea [24] with no sign or some signs of dehydration

ii.  Received written informed consent from parents

#### Exclusion criteria.

i.  Children with bloody diarrhea, severe diseases (severe sepsis, meningitis, severe pneumonia with respiratory distress requiring intensive care and ancillary support such as oxygen inhalation, oro-pharyngeal suction etc.)

ii.  The child is in a situation that could interfere with the optimal participation to the study or constitute a particular risk of non-compliance

iii.  Currently participating in another clinical trial, and

iv.  Parents refused to give informed written consent

### Study setting

We conducted this study in a rural tertiary facility in Bangladesh located nearly sixty kilometer northwest of Dhaka, the capital city of Bangladesh. Kumudini Women's Medical College Hospital, the tertiary level sentinel health facility, is one of the oldest and largest tertiary level

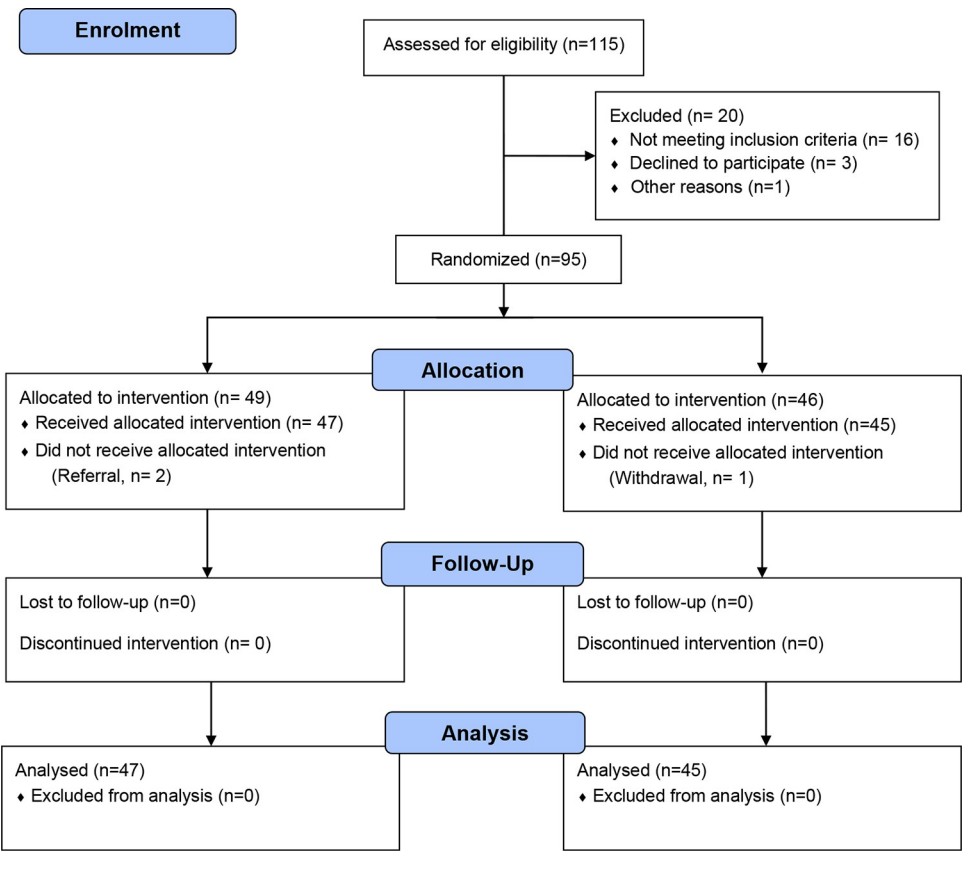

**Fig 1. Trial profile.**

facilities in rural Bangladesh having 750 beds. We conducted this study in rural Bangladesh which represents nearly 80% of our total national population.

## Intervention

Fibersol-2 (digestion resistant maltodextrin) or placebo (regular maltodextrin) dissolved in 50-ml drinking water were given orally to the study children twice daily in the morning and evening (10 gm per day) for 7 days.

## Outcomes

Outcome measures were duration of resolution of watery diarrhea, daily stool output and the proportion of children recovered within 72 hours.

## Sample size determination

Based on the results of a previous clinical trial on modified ORS solution in children with watery diarrhea [12], we anticipated that there would be a 25% relative reduction in 48-hour stool output after receiving Fibersol-2, compared to placebo. Considering 5% level of significance, 90% power and 10% drop out, the sample size was estimated to be 46 in each group. Thus, on the basis of the reduction in stool output, the total sample size was estimated to be 92.

## Randomization and masking

An experienced researcher from icddr,b, not involved in this study, prepared the randomization list using the randomization table [25]. The same independent researcher ensured the blinding by naming the study drugs 'A' and 'B' and the study personnel did not know the identity of 'A' and 'B'. The sequential randomization of 'A' and 'B' was indicated on a slip of paper, kept inside the sealed envelope. The independent researcher involved in the process wrote the sequential study number outside those envelopes. The sealed envelopes were kept under lock and key by the research team. After obtaining the informed written consent from the parents or caregivers the research physician opened the sealed envelopes in front of a non-study health worker as per randomization list and allocated the concealment as 'A' or 'B'. Thus, the interventions were masked for both the study investigators and the caregivers.

At the end of the study it is revealed that study group 'B' belongs to Fibersol-2 and study group 'A' belongs to placebo. Study group 'B', received Fibersol-2 (5 gm) dissolved in 50 ml drinking water, twice daily (suitable dose from the tolerability and acceptability trial of phase 1). Study group 'A', received placebo (5 gm of regular maltodextrin) dissolved in 50 ml drinking water twice daily (same dose of Fibersol-2).

## Patient management

If any child vomited within 10 minutes of oral intake, we repeated the similar dose for consumption after an hour of rejection. If the child vomited again within 10 minutes of previous intake, we stopped giving the intervention further. A structured questionnaire was administered at the time of enrollment to mothers to collect information on demographics, socio-economic context, clinical features and antimicrobial use prior to hospitalization and during hospitalization. Study physicians took detailed medical history of the enrolled children to determine the duration and type of diarrhea and stool frequency. Physical examination was conducted on enrollment and on the last follow-up day of 7-day observation period. Nutritional status of the study children was also measured using the standard procedures. Dehydration was assessed according to the WHO guidelines followed in the hospital [26]. In children with some dehydration, the fluid deficit was corrected with ORS in an amount 10 ml/kg/hour for the first hour, then 5 ml/kg/hour until the deficit was corrected. In addition, ongoing stool losses were replaced with ORS 5–10 ml/kg after each watery stool. For high purging children, the ORS intake was adjusted according to the ongoing stool loss. During discharge from hospital caregivers were instructed to give their child ORS @ 1 tsf/kg per loose stool, if any, during the rest of 7-days intervention period at home. Mothers were advised to continue breastfeeding at home. In addition to collect information from the hospital, our field research staff visited the households of study children to assemble information about their health status by administering the field-tested questionnaire.

Our study staff member kept observing and monitoring vital clinical parameters of the study participants at periodic intervals round the clock (24 hours) during their hospital stay. To ensure proper patient care, they maintained a roster duty with staff members. Study physicians were responsible for consenting as well as clinical assessments related to their health status, including improvements of their dehydration status and provided treatment. Apart from that they also provided treatment for other illness, when required. Study nurses were accountable for keeping vital signs and providing study drug to the children in front of their parents or caregivers at hospital and community with appropriate dose and on schedule time. They also closely monitored the participants for any unwanted events and informed study physicians.

The study consisted of 7-days intervention period followed by observation period of 7 days (in total 14 days). We discharged the study children from the hospital as soon as dehydration

was corrected and there was improvement of the concurrent illness. Sometimes on request we allowed the parents or caregivers to take away their children from hospital before complete recovery from diarrhea, if the child was playful and there was no dehydration. Our study nurses along with field organizers visited the household of the study participants and monitored the vital signs of the study children twice daily. They also measured the stool output every 8 hours interval at home during the intervention period of 7 days and once daily during the observation period for next 7 days.

## Measurements

**Fluid intake.** ORS was given to the study participants after measuring with a calibrated cylinder and the amount of intake was recorded 8 hours interval.

**Output (stool, urine, and vomitus).** In the hospital, stool was collected in a bucket of known weight beneath the cholera cot with a central hole and was measured at every 8 hours interval with an electronic scale having a precision of 1 gm. Urine was collected by a pediatric urine collector bag and measured with a calibrated cylinder in ml. Vomitus was collected in a pre-weighted bowl and measured with an electronic scale in gram. Clinical evaluation was performed in every morning and evening. Resolution of diarrhea was defined as the passage of two consecutive soft/formed stools or no stool for last 24 hours. Caregivers collected stool and urine in poly bag every 8 hours interval as per direction from the research team at home. Our health workers measured the stool and urine output every 8 hours interval after being collected by caregivers at home. Therapeutic success was defined as the cessation of diarrhea within 7 days of inclusion in the study intervention. Duration of diarrhea was calculated in hours from the time of randomization to the last watery or loose stool within 7 days. Children were considered withdrawn from the study when their parents or legal guardian withdrew consent, or the child was withdrawn from the study for the treatment of any complications. Data (intake and output) of such children was not included in the analysis (per protocol analysis).

## Operational definitions

**Acute watery diarrhea.** Passage of three or more abnormally loose or watery stool in the last 24 hours.

**Resolution of diarrhea.** Defined as the passage of two consecutive soft/formed stools or no stool for the last 24 hours.

**Abdominal distension.** Accumulation of air (gas) or fluid in the abdominal cavity causing its outward expansion beyond the normal girth of the stomach and waist. Distension is the objective enlargement of the abdomen.

**Rumbling.** Noise produced by the movement of the gastrointestinal contents while propelling through the small intestine by a series of muscle contractions. It was measured through history taking from the parents and by performing abdominal auscultation.

**Bloating.** Bloating is the presence of abnormal general swelling or increase in the diameter of the abdominal area. It was also evaluated by regular measurement of abdominal girth. It is the subjective feeling that the abdomen feels more fuller than it should be, but does not necessarily mean that the abdomen is enlarged.

**Stool consistency.** Physical form of the stool, such as solid, paste, loose/watery, identified by the parents as well as treating physicians.

The consistency of each stool was assessed and recorded as follows:

- Type 1 (Formed) = Stool having its own shape.

- Type 2 (Soft) = Stool that can't be poured, but takes the shape of a container.

• Type 3 (Watery/liquid) = Stool that can be spilled from one container to another.

Type 1 and type 2 stools were considered normal, not constitute as diarrhea. Type 3 stool was usual in bacterial/viral diarrhea.

### Statistical analysis

All the data were entered into SPSS for Windows (version 20.0; SPSS Inc., Chicago) and Epi-Info (version 7.0, USD, Stone Mountain, GA). For categorical variables, differences in proportions were compared by the Chi-square/Fisher's exact test and represented by frequency with percentage. For normally distributed continuous variables, differences of mean were compared by the Student's *t*-test and represented by mean with standard deviation (SD); the Mann-Whitney test was used for comparison of differences for those continuous variables that were not normally distributed and represented by median with inter quartile range (IQR) (Tables 1 and 2). For continuous outcome variables, we calculated unadjusted and adjusted mean differences with 95% confidence intervals (CIs) and adjusted effect sizes by multivariable linear regression model (Table 3). For categorical outcome variable, we calculated unadjusted and adjusted odds ratios (OR) by multivariable binary logistic regression model. We also calculated adjusted risk differences (Table 4). Additionally, we evaluated OR, CIs and p-values between placebo and intervention groups for abdominal symptoms in different intervention days with doses (Table 5). Statistical significance for all analyses was taken at 5% level ($p < 0.05$).

## Results

Distribution of gender, family size and household wealth quintiles were comparable between the groups. Other indicators of socio-economic-demographic characteristics, like maternal and paternal illiteracy, number of sleeping rooms, use of non-sanitary facilities and use of deep tube well water for domestic purposes were also comparable between the groups (Table 1). Diarrheal children who received placebo or Fibersol-2 were similar at enrollment in regards to age, mid upper arm circumference (MUAC), height/length, body weight, length/height-for-age z-score, weight-for-length/height z-score, and weight-for-age z-score, respectively. Distribution of underweight, wasting, and stunting was comparable between the groups (Table 1). Throughout the intervention period the volume of ORS intake was 1230 (700, 1970) ml vs. 1380 (870, 2225) ml among the children who received placebo and Fibersol-2, respectively (Table 2). The amount of vomitus, urine and stool output in hospital and at home in both the groups was also comparable (Table 2). There was no significant difference observed in terms of duration of resolution of diarrhea (adjusted mean difference 8.20, 95% CI -2.74 to 19.15, p = 0.14, adjusted effect size 0.03); the daily stool output (adjusted mean difference 73.57, 95% CI -94.17 to 241.32, p = 0.38, adjusted effect size 0.33) (Table 3) and the proportion of children recovered within 72 hours (adjusted OR: 0.49, 95% CI = 0.12 to 1.96, p = 0.31, adjusted risk difference -0.06 (95% CI -0.19 to -0.06), after regression analysis between Fibersol-2 and placebo (Table 4).

No significant difference was observed in terms of development of gastrointestinal symptoms, such as abdominal pain, distension, rumbling and bloating during the intervention period of 7 days between the two groups (Table 5).

### Safety of the trial

We evaluated same number of healthy and diarrheal children (30 each) 1–3 years old for the digestive tolerability and acceptability of Fibersol-2 before initiation of the randomized trial, where we found the product was safe and well tolerated based on disappearance or

**Table 1. Baseline characteristics.**

| Characteristics | Placebo (n = 47) | Fibersol-2 (n = 45) |
|---|---|---|
| Age in months (median, IQR) | 15 (13, 21) | 14 (12, 21) |
| Male gender | 29 (62) | 28 (62) |
| Family size (mean, SD) | 5.02 (1.54) | 5.60 (1.96) |
| Number of sleeping rooms (mean, SD) | 2.15 (0.98) | 2.33 (1.13) |
| Maternal illiteracy | 1 (2) | 1 (2) |
| Paternal illiteracy | 3 (6) | 6 (13) |
| Use of non-sanitary facility | 11 (23) | 9 (20) |
| Deep tube well | 39 (83) | 39 (86) |
| | | |
| **Wealth quintile** | | |
| Rich | 10 (21.3) | 8 (17.8) |
| Upper middle | 9 (19.1) | 10 (22.2) |
| Middle | 12 (25.5) | 6 (13.3) |
| Lower middle | 7 (14.9) | 12 (26.7) |
| Poor | 9 (19.1) | 9 (20.0) |
| | | |
| **Domestic livestock** | | |
| Cow / Buffalo | 18 (38) | 15 (33) |
| Goat | 5 (10) | 4 (9) |
| Chicken / Duck | 23 (49) | 22 (49) |
| Pigeon | 3 (6) | 5 (11) |
| Cat | 46 (98) | 43 (96) |
| Dog | 44 (94) | 45 (100) |
| | | |
| **Anthropometry** <br> Height in cm (mean, SD) | 78.61 (4.95) | 78.79 (6.23) |
| Weight in kg (mean, SD) | 9.73 (1.47) | 9.82 (1.66) |
| MUAC in cm (mean, SD) | 14.97 (1.45) | 14.74 (1.09) |
| | | |
| **Nutritional status** | | |
| Height-for-age z-score (mean, SD) | -0.58 (1.19) | -0.51 (1.07) |
| Weight-for-length/height z-score (mean, SD) | -0.53 (1.20) | -0.47 (1.14) |
| Weight-for-age z-score (mean, SD) | -0.67 (1.19) | -0.59 (1.03) |
| Stunting | 3 (6) | 3 (7) |
| Underweight | 7 (15) | 2 (4) |
| Wasting | 7 (15) | 2 (4) |

improvements in abdominal symptoms, like distension, pain, rumbling and bloating in both groups of children (journal.pone.0274302), henceforth these were the variables selected to monitor the safety of the trial.

## Discussion

In our randomized double-blind placebo-controlled parallel trial we aimed to evaluate the efficacy of Fibersol-2 along with ORS treatment to reduce the duration of watery diarrhea and stool output in children of 1–3 years as well as recovery of children within 72 hours where no beneficial role of Fibersol-2 was seen in reducing the outcomes, compared to placebo. The mean duration of resolution of diarrhea from the enrollment was comparable between placebo and Fibersol-2. All of the study participants recovered from watery stool within 7-day

**Table 2. Volume of ORS intake, amount of vomitus, urine output and stool output during the intervention period of 7 days.**

| Variables | Placebo (n = 47) | Fibersol-2 (n = 45) |
|---|---|---|
| ORS intake in ml (mean, SD) (median, IQR) | 1690.43 (1528.25) 1230 (700, 1970) | 2015.80 (2108.40) 1380 (870, 2225) |
| *Amount of vomitus in gram (mean, SD) | 66.83 (120.15) | 38.96 (60.85) |
| Urine output in ml (mean, SD) (median, IQR) | 2481.04 (282.07) 2460 (2305, 2630) | 2429.91 (337.95) 2406 (2288, 2640) |
| Stool output in gram in hospital (mean, SD) Stool output in gram at home (mean, SD) | 701.85 (505.40) 393.46 (208.50) | 898.42 (834.78) 424.51 (280.46) |

Values were expressed as mean (SD), (median, IQR); SD, Standard deviation; IQR, Inter quartile range; ORS, Oral rehydration solution.

* Only 21 children in group A and 19 children in group B had vomiting.

**Table 3. Regression analysis for continuous outcomes.**

| Characteristics | Fibersol-2 | Placebo | Unadjusted difference 95% CI | p-value | Adjusted difference 95% CI | p-value | Adjusted effect size |
|---|---|---|---|---|---|---|---|
| Duration of resolution of diarrhea mean (SE) | 44.15 (3.65) | 35.68 (3.98) | 8.47 (-2.28 to 19.23) | 0.121 | 8.20 (-2.74 to 19.15) | 0.140 | 0.03 |
| Daily stool output mean (SE) | 794.64 (82.40) | 680.0 (59.20) | 114.60 (-85.66 to 314.86) | 0.259 | 73.57 (-94.17 to 241.32) | 0.386 | 0.33 |

Data were mean (SE) unless otherwise stated. SE, Standard Error.

**Table 4. Regression analysis for categorical outcome.**

| Characteristics | Fibersol-2 | Placebo | Unadjusted odds ratio 95% CI | p-value | Adjusted odds ratio 95% CI | p-value | Adjusted risk difference (95% CI) |
|---|---|---|---|---|---|---|---|
| Patients recovered within 72 hours | 37 (82) | 43 (91) | 0.43 (0.12 to 1.54) | 0.196 | 0.49 (0.12 to 1.96) | 0.313 | -0.06 (-0.19 to -0.06 |

Data were n (%), unless stated otherwise.

intervention period in the study. We do not have any ready reference to explain such a result other than relatively small sample size that caused reduced power of the study. There was the longer trend of diarrheal duration on enrollment observed in study children received Fibersol-2 compared to placebo. Interestingly, the result showed that the diarrheal duration on enrollment and the duration to recover from diarrhea were very similar for both groups. Therefore, the differences in diarrhea duration between groups on enrollment may have affected the outcomes, and the effectiveness of Fibersol-2 was not shown in the present study.

It is our speculation that the environmental changes such as climate change, home environment, household pets, the use of non-sanitary facility may affect the duration of diarrheal improvement. But as our study is a randomized controlled trial, the effects of environmental changes should be upon both group of children and it might not cause any change in the overall outcome of the study.

**Table 5. Characteristics of gastrointestinal symptoms after ingestion of study drugs during the intervention period in both the groups.**

| Characteristics | Placebo (n = 47) | Fibersol-2 (n = 45) | OR (95% CI) | p-value |
|---|---|---|---|---|
| **At day 1, 1st dose** | | | | |
| Presence of abdominal distension | 1 (2) | 0 (0) | - | 0.511 |
| Presence of abdominal pain | 1 (2) | 2 (4) | 0.48 (0.05–5.09) | 0.484 |
| Presence of abdominal rumbling | 3 (6) | 1 (2) | 2.87 (0.31–26.60) | 0.325 |
| Presence of abdominal bloating | 6 (13) | 3 (7) | 1.92 (0.51–7.20) | 0.265 |
| **At day 1, 2nd dose** | | | | |
| Presence of abdominal distension | 0 (0) | 2 (4) | - | 0.237 |
| Presence of abdominal pain | 1 (2) | 1 (2) | 0.96 (0.06–14.85) | 0.742 |
| Presence of abdominal rumbling | 1 (2) | 1 (2) | 0.96 (0.06–14.85) | 0.742 |
| Presence of abdominal bloating | 2 (4) | 2 (4) | 0.96 (0.14–6.51) | 0.675 |
| **At day 2, 1st dose** | | | | |
| Presence of abdominal distension | 0 (0) | 3 (7) | - | 0.113 |
| Presence of abdominal pain | 1 (0) | 0 (0) | - | 0.511 |
| Presence of abdominal rumbling | 1 (2) | 1 (2) | 0.96 (0.06–14.85) | 0.742 |
| Presence of abdominal bloating | 1 (2) | 4 (9) | 0.24 (0.03–2.06) | 0.167 |
| **At day 2, 2nd dose** | | | | |
| Presence of abdominal distension | 0 (0) | 1 (2) | - | 0.489 |
| Presence of abdominal pain | 0 (0) | 1 (2) | - | 0.489 |
| Presence of abdominal rumbling | 2 (4) | 1 (2) | 1.92 (0.18–20.39) | 0.516 |
| Presence of abdominal bloating | 3 (6) | 2 (4) | 1.44 (0.25–8.20) | 0.521 |
| **At day 3, 1st dose** | | | | |
| Presence of abdominal distension | 1 (2) | 0 (0) | - | 0.511 |
| Presence of abdominal pain | 0 (0) | 2 (4) | - | 0.237 |
| Presence of abdominal rumbling | 2 (4) | 1 (2) | 1.92 (0.18–20.39) | 0.516 |
| Presence of abdominal bloating | 1 (2) | 2 (4) | 0.48 (0.05–5.09) | 0.484 |
| **At day 3, 2nd dose** | | | | |
| Presence of abdominal distension | 0 (0) | 0 (0) | - | - |
| Presence of abdominal pain | 1 (2) | 1 (2) | 0.96 (0.06–14.85) | 0.742 |
| Presence of abdominal rumbling | 3 (6) | 1 (2) | 2.87 (0.31–26.60) | 0.325 |
| Presence of abdominal bloating | 1 (2) | 1 (2) | 0.96 (0.06–14.85) | 0.742 |
| **At day 4, 1st dose** | | | | |
| Presence of abdominal distension | 0 (0) | 0 (0) | - | - |
| Presence of abdominal pain | 0 (0) | 0 (0) | - | - |
| Presence of abdominal rumbling | 2 (4) | 1 (2) | 1.92 (0.18–20.39) | 0.516 |

*(Continued)*

**Table 5.** (Continued)

| Characteristics | Placebo (n = 47) | Fibersol-2 (n = 45) | OR (95% CI) | p-value |
|---|---|---|---|---|
| Presence of abdominal bloating | 2 (4) | 1 (2) | 1.92 (0.18–20.39) | 0.516 |
| **At day 4, 2nd dose** | | | | |
| Presence of abdominal distension | 0 (0) | 0 (0) | - | - |
| Presence of abdominal pain | 0 (0) | 0 (0) | - | - |
| Presence of abdominal rumbling | 2 (4) | 0 (0) | - | 0.258 |
| Presence of abdominal bloating | 2 (4) | 2 (4) | 0.96 (0.14–6.51) | 0.675 |
| **At day 5, 1st dose** | | | | |
| Presence of abdominal distension | 0 (0) | 0 (0) | - | - |
| Presence of abdominal pain | 0 (0) | 0 (0) | - | - |
| Presence of abdominal rumbling | 1 (2) | 2 (4) | 0.48 (0.05–5.09) | 0.484 |
| Presence of abdominal bloating | 1 (2) | 1 (2) | 0.96 (0.06–14.85) | 0.742 |
| **At day 5, 2nd dose** | | | | |
| Presence of abdominal distension | 0 (0) | 1 (2) | - | 0.489 |
| Presence of abdominal pain | 0 (0) | 1 (2) | - | 0.489 |
| Presence of abdominal rumbling | 0 (0) | 1 (2) | - | 0.489 |
| Presence of abdominal bloating | 1 (2) | 2 (4) | 0.48 (0.05–5.09) | 0.484 |
| **At day 6, 1st dose** | | | | |
| Presence of abdominal distension | 0 (0) | 1 (2) | - | 0.489 |
| Presence of abdominal pain | 0 (0) | 0 (0) | - | - |
| Presence of abdominal rumbling | 0 (0) | 0 (0) | - | - |
| Presence of abdominal bloating | 1 (2) | 2 (4) | 0.48 (0.05–5.09) | 0.484 |
| **At day 6, 2nd dose** | | | | |
| Presence of abdominal distension | 0 (0) | 0 (0) | - | - |
| Presence of abdominal pain | 0 (0) | 0 (0) | - | - |
| Presence of abdominal rumbling | 0 (0) | 0 (0) | - | - |
| Presence of abdominal bloating | 0 (0) | 0 (0) | - | - |
| **At day 7, 1st dose** | | | | |
| Presence of abdominal distension | 0 (0) | 2 (4) | - | 0.237 |
| Presence of abdominal pain | 0 (0) | 0 (0) | - | - |
| Presence of abdominal rumbling | 1 (2) | 0 (0) | - | 0.511 |
| Presence of abdominal bloating | 0 (0) | 2 (4) | - | 0.237 |
| **At day 7, 2nd dose** | | | | |
| Presence of abdominal distension | 0 (0) | 0 (0) | - | - |
| Presence of abdominal pain | 0 (0) | 1 (2) | - | 0.489 |
| Presence of abdominal rumbling | 0 (0) | 0 (0) | - | - |
| Presence of abdominal bloating | 0 (0) | 0 (0) | - | - |

Values were expressed as number of subjects as n; p-value was assessed by t-test or Chi-square test.

Slow fermentation pattern in large intestine is one of characteristics of Fibersol-2 among various fermentable soluble dietary fibers. A study conducted in humans showed slower fermentation rate of Fibersol-2 compared to other soluble non digestible carbohydrates such as

fructo oligosaccharides, guar gum, and PHGG [27]. Fibersol-2 provides source of fermentable carbohydrates to the distal part of large intestine resulting in production of SCFAs throughout the intestine and facilitates absorption of sodium chloride and water in the colon thus help in forming solid stool. They serve as an essential energy source for the colonocytes and also excite epithelial cell proliferation, thus SCFAs help in mucosal protection as well as in building mucosal lining in both large and small gut [28]. A study conducted by Alam NH *et al.* in Bangladesh suggested that PHGG supplemented in ORS and other unabsorbed carbohydrates can appreciably help in improving the health condition of severely malnourished young diarrheal children by reducing duration of diarrhea and decreasing stool output. However, the study didn't observe any significant reduction in stool weight nor explain specific advantageous role of PHGG in severely malnourished children [12]. In that particular study PHGG was supplemented in ORS, while in the present study Fibersol-2 was dissolved in 50-ml drinking water and administered separately from ORS.

One of the major limitations of this study was accuracy in collecting stool from the female children. Despite several strict attempts to measure the stool volume accurately, mixing of urine particularly in case of female children with the stool by chance might have taken place which made the stool amount more than actual. Moreover, the study did not examine the presence of bacterial or viral diarrheagenic pathogens, thus, the study failed to identify the etiologic differentials in diarrheal duration and stool output. Potential inflation of a type I error may occur as a result of multiple testing of subgroup comparisons, treatment arms, outcomes and analyses of the same outcome at different times. This can be prevented by complete and accurate reporting of the analyses being outlined in the registered trial protocols and mitigated by various statistical adjustment methods.

## Conclusions

Although, Fibersol-2 was found to be digestively well-tolerable and safe in children, the study observed no beneficial role of Fibersol-2 in reducing the duration of watery diarrhea and stool output as well as recovery of children within 72 hours, compared to placebo. Further studies with large sample are imperative to refute or accept our observations.

## Supporting information

**S1 Checklist. CONSORT 2010 checklist of information to include when reporting a randomised trial**\*.
(DOCX)

**S1 Appendix.**
(DOCX)

**S1 File. Study Questionnaire English and Bengali.**
(PDF)

**S2 File. Trial study protocol.**
(DOCX)

## Acknowledgments

We would like to express our sincere thanks to all research physicians, nurses, other research and hospital staff for their invaluable support and contribution during patient enrollment and data collection. We would like to express our gratitude to care-givers/mothers of the study participants for their consent to enroll their children in the study.

## Author Contributions

**Conceptualization:** Abu Sadat Mohammad Sayeem Bin Shahid, Shahnawaz Ahmed, Mohammod Jobayer Chisti.

**Data curation:** Abu Sadat Mohammad Sayeem Bin Shahid, Shahnawaz Ahmed, Sampa Dash, Mohammod Jobayer Chisti.

**Formal analysis:** Abu Sadat Mohammad Sayeem Bin Shahid, Shahnawaz Ahmed, Mohammod Jobayer Chisti.

**Funding acquisition:** Abu Sadat Mohammad Sayeem Bin Shahid, Shahnawaz Ahmed, Yuka Kishimoto, Sumiko Kanahori, Abu Syed Golam Faruque, Mohammod Jobayer Chisti.

**Investigation:** Abu Sadat Mohammad Sayeem Bin Shahid, Shahnawaz Ahmed, Mohammod Jobayer Chisti.

**Methodology:** Abu Sadat Mohammad Sayeem Bin Shahid, Mohammod Jobayer Chisti.

**Project administration:** Abu Sadat Mohammad Sayeem Bin Shahid, Shahnawaz Ahmed, Mohammod Jobayer Chisti.

**Resources:** Abu Sadat Mohammad Sayeem Bin Shahid, Mohammod Jobayer Chisti.

**Software:** Abu Sadat Mohammad Sayeem Bin Shahid, Shahnawaz Ahmed, Mohammod Jobayer Chisti.

**Supervision:** Abu Sadat Mohammad Sayeem Bin Shahid, Shahnawaz Ahmed, Mohammod Jobayer Chisti.

**Validation:** Abu Sadat Mohammad Sayeem Bin Shahid, Shahnawaz Ahmed, Sampa Dash, Yuka Kishimoto, Sumiko Kanahori, Tahmeed Ahmed, Abu Syed Golam Faruque, Mohammod Jobayer Chisti.

**Visualization:** Abu Sadat Mohammad Sayeem Bin Shahid, Shahnawaz Ahmed, Mohammod Jobayer Chisti.

**Writing – original draft:** Abu Sadat Mohammad Sayeem Bin Shahid.

**Writing – review & editing:** Shahnawaz Ahmed, Sampa Dash, Yuka Kishimoto, Sumiko Kanahori, Tahmeed Ahmed, Abu Syed Golam Faruque, Mohammod Jobayer Chisti.

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
