## [Decision Letter · Decision Letter 0]

3 Mar 2021

PONE-D-21-01310

Does Fibersol-2 efficacious in reducing duration of watery diarrhea and stool output in children 1-3 years old? a randomized, parallel, double-blinded, placebo- controlled, two arm clinical trial

PLOS ONE

Dear Dr. Shahid,

Thank you for submitting your manuscript to PLOS ONE. After careful consideration, we feel that it has merit but does not fully meet PLOS ONE’s publication criteria as it currently stands. Therefore, we invite you to submit a revised version of the manuscript that addresses the points raised during the review process.

Your manuscript has been reviewed by three experts in the field, and they have found some points that need to be addressed before this manuscript is considered for publication. Please go through the reviewers' comments and consider addressing these points, and prepare a revised version.

We look forward to receiving your revised manuscript.

Kind regards,

Ivan D. Florez; MD, MSc, PhD

Academic Editor

PLOS ONE

Journal Requirements:

3. Please provide a sample size and power calculation in the Methods, or discuss the reasons for not performing one before study initiation.

4. During your revisions, please confirm whether the wording in the title is correct and update it in the manuscript file and online submission information if needed. Specifically, we suggest that "Does" should be changed to "Is", and the "a" after the question mark should be capitalised ("A").

5. Thank you for submitting your clinical trial to PLOS ONE and for providing the name of the registry and the registration number. The information in the registry entry suggests that your trial was registered after patient recruitment began. PLOS ONE strongly encourages authors to register all trials before recruiting the first participant in a study.

1) your reasons for your delay in registering this study (after enrolment of participants started);

2) confirmation that all related trials are registered by stating: “The authors confirm that all ongoing and related trials for this drug/intervention are registered".

6. We note that you have indicated that data from this study are available upon request. PLOS only allows data to be available upon request if there are legal or ethical restrictions on sharing data publicly. For information on unacceptable data access restrictions, please see http://journals.plos.org/plosone/s/data-availability#loc-unacceptable-data-access-restrictions.

7. We note you have included a table to which you do not refer in the text of your manuscript. Please ensure that you refer to Table 4 and 5 in your text; if accepted, production will need this reference to link the reader to the Table.

8. Thank you for stating the following in the Financial Disclosure:

"This research study was funded by Matsutani Chemical Industry Company Limited, Japan on behalf of ADM/Matsutani LLC, USA. The International Centre for Diarrhoeal Disease Research, Bangladesh, receives unrestricted support from the Government of the People's Republic of Bangladesh, Global Affairs Canada, the Swedish International Development Cooperation Agency and the UK Department for International Development."

We note that one or more of the authors have an affiliation to the commercial funders of this research study : Matsutani Chemical Industry Co.Ltd

(2) Please also provide an updated Competing Interests Statement declaring this commercial affiliation along with any other relevant declarations relating to employment, consultancy, patents, products in development, or marketed products, etc.  

9. Please include your tables as part of your main manuscript and remove the individual files. Please note that supplementary tables (should remain/ be uploaded) as separate "supporting information" files

Additional Editor Comments:

Your manuscript has been reviewed by three experts in the field, and they have found some points that need to be addressed before this manuscript is considered for publication. Please go through the reviewers' comments and consider addressing these points, and prepare a revised version.

Reviewers' comments:

Reviewer's Responses to Questions

**Comments to the Author**

1. Is the manuscript technically sound, and do the data support the conclusions?

Reviewer #1: Partly

Reviewer #2: Yes

Reviewer #3: No

2. Has the statistical analysis been performed appropriately and rigorously? 

Reviewer #1: Yes

Reviewer #2: No

Reviewer #3: Yes

3. Have the authors made all data underlying the findings in their manuscript fully available?

Reviewer #1: No

Reviewer #2: No

Reviewer #3: Yes

4. Is the manuscript presented in an intelligible fashion and written in standard English?

Reviewer #1: No

Reviewer #2: Yes

Reviewer #3: No

5. Review Comments to the Author

Reviewer #1: 1. On the assessment of patient´s response: In the introduction section, the authors mention in a rather casual way that they wanted to study this compound's effect on the duration of diarrhea and the stool output (lines 88-89 of the manuscript). In another paragraph, under the "measurements" section (lines 201- 202), the authors indicate that the primary outcomes are diarrhea duration, the proportion of patients recovered within 72 hours, and daily stool output. Therefore, this clinical trial has not one but three primary criteria of response. To measure diarrhea duration, the authors indicate that they did count the number of hours elapsed from randomization to the last watery stool (lines 196-197). The authors need to clarify if they were recording the physical characteristics of every single stool passed during the study. It looks to me that only the stool volume was measured and recorded. This clarification will also be useful for understanding how they calculated the proportion of patients recovered from diarrhea. Enrolled patients stayed in the hospital for some days and then discharged and followed up at home. This design introduces a potential major problem in measuring the stool volume. While in the hospital, stools were collected in buckets placed beneath a cholera cot (line 186); once the patients were at home, the stools were collected in plastic bags (lines 192-193). These are two different ways of measuring this study outcome. At best, the authors should calculate the stool output during the hospital stay separated from that collected at home and compare this variable between the two groups looking for inconsistencies.

2. On the required size of study: The authors do not provide a sample size calculation, which should include the study power and the Type I and Type II error levels for any of the three selected study outcomes; they only mention the number of subjects included in the trial (line 108). I suspect that the necessary number of subjects to ascertain the study hypothesis would differ for each study outcome since each uses a different metric (hours, proportion, and volume).

3. On the method used to prepare a list of random treatment assignements: The authors inform that this is a placebo-controlled, randomized, double-blind, parallel clinical trial (line 105). A person not involved otherwise in the study prepared the list of random treatment assignments using a random number table (lines 114-115). The treatment's name was written on a piece of paper and then placed inside a sealed envelope (lines 115-116). The authors must clarify whether the person who prepared the sealed envelopes wrote the sequential study number outside these envelopes.

4. On the allocation concealment: The study researcher had the envelopes containing the treatment assignments (line 117) and opened the correspondent envelope only after the next patient was enrolled and gave the consent (lines 118-120). However, the authors need to explain the procedure followed in greater detail.

5. On the double-blind nature of this study: This is not a double-blind clinical trial. In this study, the enrolled patients received either the experimental treatment (digestion resistant maltodextrin) or the control treatment (regular maltodextrin) in the form of a drinking solution, twice a day for seven days (lines 109-112). The study was not blinded for the research physician since he or she was in charge of opening the sealed envelopes containing the name of the treatment allocated to the next patient (lines 118-119). We do not know if the study was blinded to the caregivers and how. The authors need to clarify this. The authors also should explain why they did not opt for a double-blind design, which would be feasible if the sequential study drinking solutions were provided already prepared by the hospital's pharmacy labeled only with the sequential study number.

6. On the statistical analysis: Mean (standard deviation), and median (IQR) for continuous variables and OR (95% CI) for categorical variables are provided in the tables presenting the data analysis results. Since this study is negative (no superiority of the experimental treatment over the control treatment), the authors must estimate how large the type II error is for this study's size.

7. On the evaluation of safety: There is no section in the manuscript, neither in methods or results, that explicitly presents what and how to monitor the intervention's side effects. Table 5 shows the comparative results between the two treatment groups of four variables observed during the seven days of the study: abdominal distension, abdominal pain, abdominal rumbling, and abdominal bloating. The authors need to clarify if these were the variables selected to monitor the safety of the trial. I do not understand the difference between abdominal distension and abdominal bloating.

8. On the registration of this trial: I have verified that this trial is registered at ClinicalTrials.gov with the number NCT03565393

9. On the accessibility of the study protocol and primary data records: The study protocol is available as an annex (supporting information). The authors declared that this study's primary data records would be public with some restrictions and only upon request.

10. On the sources of funding and the role of funders: The sponsor of this study is mentioned in the manuscript. However, the authors do not note whether or not the funder had a role in designing the study protocol, data analysis, decision to publish, and preparing this manuscript.

Reviewer #2: Title: There's a typo in the title; did you intend for it to be; "Is Fibersol-2 efficacious..."

Abstract: The presentation of the numerical results should be improved, for example for the continuous outcomes you should report the mean, standard error, difference, 95% confidence intervals for the difference and p-value: "The mean (SE) duration of watery diarrhoea in the fibserol and control groups respectively were XX(xx) and YY(yy) respectively, a difference of ZZ, 95%CI zz to zz, p-value zzzz." For the binary outcome: "The proportions of children who recovered from watery diarrhoea within 72 hours in the fibersol and control arms were AA and BB respectively, odds/risk ratio/difference CC, 95%CI cc to cc, p-value cccc."

Introduction:

- there's a missing space between "fibre" and "produced" in line 60.

- first word of line 61 should be plural "prebiotics", or begin with "A prebiotic..."

- line 70: "the evidence", not plural.

- line 78: do you mean "in children admitted to hospital with diarrhoea and dehydration" or "dehydration due to diarrhoea", rather than "dehydrated diarrhoea"?

- line 84, you mean "healthy" not "healthful".

Methods

Although most of the required information is present, the reporting of the methods does not follow the order of sections recommended in the CONSORT guidelines, and the methods therefore feels rather jumbled up. For example, lines 108 to 112 under 'study design' belong to 'randomisation and masking', and the description of the eligibility criteria after randomisation and masking seems odd. Please have a look at the CONSORT statement at http://www.consort-statement.org/checklists/view/32--consort-2010/66-title and try to reorganise the methods to follow the suggested order.

Please include a clear description of how the sample size was determined - this is not currently reported. The numbers of children in each arm is a result and should not be included anywhere in the methods, e.g. in lines 121 and 123.

For the analysis, first, a table of descriptive characteristics of the sample, without any statistical tests comparing the groups, should be presented. These should be means and standard deviations for continuous variables, and counts and proportions for categorical ones, in each group and overall. This should be Table 1. It should be followed by a table showing the mean and SE of continuous outcomes in each group and the difference in means, 95% confidence intervals and p-values, both crude and adjusted, from a linear regression model. For the binary outcome the table should have the counts of events and proportions in each arm and the odds ratio or risk ratio or risk difference with the 95% confidence intervals and p-values, both crude and adjusted. It is these results that should then be summarised in the text and abstract.

Please avoid using the ± designation anywhere in the text as it implies a range of values which is not what you seem to mean in each case where it is used.

Please include the sources of funding and other support and the role of funders as a sub-section in the methods (e.g. the last subsection of methods).

Reviewer #3: Dear Editor

This study is a good example for the effects of different kind of supplementation to ORS for children with acute infectious diarrhea. Randomization, patient selection and end-points are great for pediatric diarrhea study. However I can not understand why the authors prefer to use this prebiotic for the treatment of diarrhea. Prebiotics have some beneficial effects on health (not innumerable), and majority of the effects of fiber are increased transit time (mainly proposed in children with constipationd or other FGIDs). This may be beneficial for diarrhea if combined with probiotic strains.

6. PLOS authors have the option to publish the peer review history of their article (what does this mean?). If published, this will include your full peer review and any attached files.

Reviewer #1: **Yes: **Eduardo Salazar-Lindo

Reviewer #2: No

Reviewer #3: No

---

## [Author Response · Author response to Decision Letter 0]

31 Oct 2021

Date: 31st October, 2021

To

Ivan D. Florez

From: 

Dr. Abu Sadat Mohammad Sayeem Bin Shahid

Corresponding Author

Subject: Response to the comments of academic editor and the reviewers of PLOS ONE on manuscript Ref: PONE-D-21-01310 titled “Is Fibersol-2 efficacious in reducing duration of watery diarrhea and stool output in children 1-3 years old? A randomized, parallel, double-blinded, placebo-controlled, two arm clinical trial.”

Dear Ivan D. Florez,

Thank you for evaluating our manuscript and providing us with the opportunity to submit the revised manuscript after addressing academic editor’s and reviewer’s comments. We also express our sincere thanks to them for evaluating our manuscript. We are sending a track change version as well as a clean version of the manuscript that highlights the changes we have made from the previous version. We are also attaching this letter outlining a point-by-point response to the each point kindly raised by the academic editor and respected reviewers. 

We hope that our response will be appropriate to qualify the manuscript for publication in your well-reputed journal. 

We look forward to kindly hearing from you.

Thank you.

Responses to the comments of Academic Editor and the respected reviewers

Journal additional requirements:

1. Please clarify if the following authors were employed by Matsutani Chemical Industry Co Ltd, Hyogo, Japan at the time the study took place:

 Yuka Kishimoto

 Sumiko Kanahori

 Abu Syed Golam Faruque

 Mohammod Jobayer Chisti

Please also clarify if the above authors are currently employed Matsutani Chemical Industry Co Ltd.

Response: Thank you. Yuka Kishimoto and Sumiko Kanahori were employed by Matsutani Chemical Industry Co Ltd, Hyogo, Japan at the time the study took place and contributed as co-authors of the manuscript. They currently belong to the same institution. The rest of two were affiliated by International Centre for Diarrhoeal Disease Research, Bangladesh (icddr,b) during the study period and currently belong to the same organization.

Response: Thank you. It has been revised accordingly.

Response: Thank you. It has been uploaded as Supporting Information.

3. Please provide a sample size and power calculation in the Methods, or discuss the reasons for not performing one before study initiation.

Response: Thank you for the comment. It has been incorporated in line no 141-156, page no.7-8 and line no 305-306, page no.14 in clean version of the manuscript.

4. During your revisions, please confirm whether the wording in the title is correct and update it in the manuscript file and online submission information if needed. Specifically, we suggest that "Does" should be changed to "Is", and the "a" after the question mark should be capitalised ("A").

Response: Thank you. It has been revised accordingly in line no.1-2, page no. 1

5. Thank you for submitting your clinical trial to PLOS ONE and for providing the name of the registry and the registration number. The information in the registry entry suggests that your trial was registered after patient recruitment began. PLOS ONE strongly encourages authors to register all trials before recruiting the first participant in a study.

1) your reasons for your delay in registering this study (after enrolment of participants started);

2) confirmation that all related trials are registered by stating: “The authors confirm that all ongoing and related trials for this drug/intervention are registered".

Response: Thank you for the suggestions. As per as our knowledge we started the patient enrolment just after initiation of the registration process through our research administration. However, the process of accomplishment for the registration took long time and thus, the online system showed the delay in registration. Please accept our sincere apology for such unintentional error. 

6. We note that you have indicated that data from this study are available upon request. PLOS only allows data to be available upon request if there are legal or ethical restrictions on sharing data publicly. For information on unacceptable data access restrictions, please see http://journals.plos.org/plosone/s/data-availability#loc-unacceptable-data-access-restrictions.

Response: Thank you for the comment. Our data contain a lot of personal information where we de-identified them during analysis. As per institutional policy data with personal information will be remained with our Research Administration (RA) due to ethical constraint and if someone wants to make the availability of the de-identified data, he/she may kindly communicate with the head of RA (aahmed@)icddrb.org).

7. We note you have included a table to which you do not refer in the text of your manuscript. Please ensure that you refer to Table 4 and 5 in your text; if accepted, production will need this reference to link the reader to the Table.

Response: Thank you. Number of tables has been revised in the result section of the manuscript.

8. Thank you for stating the following in the Financial Disclosure:

"This research study was funded by Matsutani Chemical Industry Company Limited, Japan on behalf of ADM/Matsutani LLC, USA. The International Centre for Diarrhoeal Disease Research, Bangladesh, receives unrestricted support from the Government of the People's Republic of Bangladesh, Global Affairs Canada, the Swedish International Development Cooperation Agency and the UK Department for International Development."

We note that one or more of the authors have an affiliation to the commercial funders of this research study: Matsutani Chemical Industry Co. Ltd

Response: Thank you for the suggestions. It has been incorporated in line no 275-276, page no.13 in the clean version of the manuscript.

(2) Please also provide an updated Competing Interests Statement declaring this commercial affiliation along with any other relevant declarations relating to employment, consultancy, patents, products in development, or marketed products, etc.

Response: Thank you for the comment. It has been incorporated in line no. 380-382, page no.18 in the clean version of the manuscript.

9. Please include your tables as part of your main manuscript and remove the individual files. Please note that supplementary tables (should remain/ be uploaded) as separate "supporting information" files.

Response: Thank you for the suggestion. It has been revised accordingly in page no. 21-27 in the clean version of the manuscript.

Additional Editor Comments: 

Your manuscript has been reviewed by three experts in the field, and they have found some points that need to be addressed before this manuscript is considered for publication. Please go through the reviewers' comments and consider addressing these points, and prepare a revised version.

Response: Thank you. 

Review Comments to the Author

Reviewer #1: 

1. On the assessment of patient´s response: In the introduction section, the authors mention in a rather casual way that they wanted to study this compound's effect on the duration of diarrhea and the stool output (lines 88-89 of the manuscript). In another paragraph, under the "measurements" section (lines 201- 202), the authors indicate that the primary outcomes are diarrhea duration, the proportion of patients recovered within 72 hours, and daily stool output. Therefore, this clinical trial has not one but three primary criteria of response. To measure diarrhea duration, the authors indicate that they did count the number of hours elapsed from randomization to the last watery stool (lines 196-197). The authors need to clarify if they were recording the physical characteristics of every single stool passed during the study. It looks to me that only the stool volume was measured and recorded. This clarification will also be useful for understanding how they calculated the proportion of patients recovered from diarrhea. Enrolled patients stayed in the hospital for some days and then discharged and followed up at home. This design introduces a potential major problem in measuring the stool volume. While in the hospital, stools were collected in buckets placed beneath a cholera cot (line 186); once the patients were at home, the stools were collected in plastic bags (lines 192-193). These are two different ways of measuring this study outcome. At best, the authors should calculate the stool output during the hospital stay separated from that collected at home and compare this variable between the two groups looking for inconsistencies.

Response: Thank you for the comments and valuable suggestions. We visibly evaluated the both the consistency and volume of the stool (data available in the supplementary tables S2 & S3 under supporting information). The investigators have wealth of experiences in evaluating the resolution of diarrhea on the basis of these defined criteria. Both at hospital (cholera cots) and home (in plastic bag) the evaluation of consistency and volume of the stool was not so difficult and all our study personnel have performed this following our set criteria. Thus, we trust that there is no discrimination of evaluating the stool output between hospitalization and stay at home (please see table. 3 in page no. 23 in the clean version of the manuscript).

2. On the required size of study: The authors do not provide a sample size calculation, which should include the study power and the Type I and Type II error levels for any of the three selected study outcomes; they only mention the number of subjects included in the trial (line 108). I suspect that the necessary number of subjects to ascertain the study hypothesis would differ for each study outcome since each uses a different metric (hours, proportion, and volume).

Response: Thank you for the comment. It has been incorporated in line no 141-156, page no.7-8 in clean version of the manuscript.

3. On the method used to prepare a list of random treatment assignments: The authors inform that this is a placebo-controlled, randomized, double-blind, parallel clinical trial (line 105). A person not involved otherwise in the study prepared the list of random treatment assignments using a random number table (lines 114-115). The treatment's name was written on a piece of paper and then placed inside a sealed envelope (lines 115-116). The authors must clarify whether the person who prepared the sealed envelopes wrote the sequential study number outside these envelopes.

Response: Thank you for the comment. It has been incorporated in line no.159-164, page no.8 in clean version of the manuscript.

4. On the allocation concealment: The study researcher had the envelopes containing the treatment assignments (line 117) and opened the correspondent envelope only after the next patient was enrolled and gave the consent (lines 118-120). However, the authors need to explain the procedure followed in greater detail.

Response: Thank you for the suggestion. It has been incorporated in line no. 159-167, page no.8 in clean version of the manuscript.

5. On the double-blind nature of this study: This is not a double-blind clinical trial. In this study, the enrolled patients received either the experimental treatment (digestion resistant maltodextrin) or the control treatment (regular maltodextrin) in the form of a drinking solution, twice a day for seven days (lines 109-112). The study was not blinded for the research physician since he or she was in charge of opening the sealed envelopes containing the name of the treatment allocated to the next patient (lines 118-119). We do not know if the study was blinded to the caregivers and how. The authors need to clarify this. The authors also should explain why they did not opt for a double-blind design, which would be feasible if the sequential study drinking solutions were provided already prepared by the hospital's pharmacy labeled only with the sequential study number.

Response: Thank you for the comment. We are really sorry to create the misunderstanding. We have now revised the randomization and masking section with better clarity (please see line no. 159-173, page no.8 in clean version of the manuscript).

6. On the statistical analysis: Mean (standard deviation), and median (IQR) for continuous variables and OR (95% CI) for categorical variables are provided in the tables presenting the data analysis results. Since this study is negative (no superiority of the experimental treatment over the control treatment), the authors must estimate how large the type II error is for this study's size.

Response: Thank you for the comment. It has been incorporated in line no. 305-306, page no.14 in clean version of the manuscript.

7. On the evaluation of safety: There is no section in the manuscript, neither in methods or results, that explicitly presents what and how to monitor the intervention's side effects. Table 5 shows the comparative results between the two treatment groups of four variables observed during the seven days of the study: abdominal distension, abdominal pain, abdominal rumbling, and abdominal bloating. The authors need to clarify if these were the variables selected to monitor the safety of the trial. I do not understand the difference between abdominal distension and abdominal bloating.

Response: Thank you for the comments. As we evaluated same number of healthy and diarrheal children (30 each) 1-3 years old for the digestive tolerability and acceptability of Fibersol-2 before initiation of the randomized trial, where we found the product is safe and well tolerated based on disappearance or improvements in abdominal symptoms, like distension, pain, rumbling and bloating in both groups of children (PONE-D-21-01326), henceforth these were the variables selected to monitor the safety of the trial (please see line no.309-314, page no.14-15 in clean version of the manuscript), although we didn’t mention about the safety of the interventions in the protocol. Sorry for the unintentional error. Although, both abdominal distension and bloating seem to be synonymous, we have defined them separately for the study purpose on the basis of the evidence (www.medicinenet.com) and have been described in line no. 243-244, page no 11 & line no. 250-251, page no. 12 in clean version of the manuscript.

8. On the registration of this trial: I have verified that this trial is registered at ClinicalTrials.gov with the number NCT03565393

Response: Thank you.

9. On the accessibility of the study protocol and primary data records: The study protocol is available as an annex (supporting information). The authors declared that this study's primary data records would be public with some restrictions and only upon request.

Response: Thank you.

10. On the sources of funding and the role of funders: The sponsor of this study is mentioned in the manuscript. However, the authors do not note whether or not the funder had a role in designing the study protocol, data analysis, decision to publish, and preparing this manuscript.

Response: Thank you for the comment. It has been incorporated in line no. 275-276, page no.13 in clean version of the manuscript.

Reviewer #2: 

Title: There's a typo in the title; did you intend for it to be; "Is Fibersol-2 efficacious..."

Response: Thank you. It has been revised accordingly in line no.1, page no.1

Abstract: 

The presentation of the numerical results should be improved, for example for the continuous outcomes you should report the mean, standard error, difference, 95% confidence intervals for the difference and p-value: "The mean (SE) duration of watery diarrhoea in the fibersol and control groups respectively were XX(xx) and YY(yy) respectively, a difference of ZZ, 95% CI zz to zz, p-value zzzz." For the binary outcome: "The proportions of children who recovered from watery diarrhoea within 72 hours in the fibersol and control arms were AA and BB respectively, odds/risk ratio/difference CC, 95% CI cc to cc, p-value cccc."

Response: Thank you for the suggestion. It has been revised accordingly in line no. 40-46, page no. 2-3 in clean version of the manuscript.

Introduction:

- There’s a missing space between "fibre" and "produced" in line 60.

Response: Thank you. It has been corrected accordingly in line no 63, page no.3 in clean version of the manuscript.

- First word of line 61 should be plural "prebiotics", or begin with "A prebiotic..."

Response: Thank you. It has been corrected accordingly in line no 64, page no.4 in clean version of the manuscript.

- Line 70: "the evidence", not plural.

Response: Thank you. It has been corrected accordingly in line no 73, page no. 4 in clean version of the manuscript.

- Line 78: do you mean "in children admitted to hospital with diarrhoea and dehydration" or "dehydration due to diarrhoea", rather than "dehydrated diarrhoea"?

Response: Thank you. It has been revised accordingly in line no. 81, page no. 4 in clean version of the manuscript.

- Line 84, you mean "healthy" not "healthful".

Response: Thank you. It has been corrected accordingly in line no. 87, page no. 4 in clean version of the manuscript.

Methods

Although most of the required information is present, the reporting of the methods does not follow the order of sections recommended in the CONSORT guidelines, and the methods therefore feels rather jumbled up. For example, lines 108 to 112 under 'study design' belong to 'randomisation and masking', and the description of the eligibility criteria after randomisation and masking seems odd. Please have a look at the CONSORT statement at http://www.consort-statement.org/checklists/view/32--consort-2010/66-title and try to reorganise the methods to follow the suggested order.

Response: Thank you for the suggestion. It has been revised accordingly in the CONSORT checklist.

Please include a clear description of how the sample size was determined - this is not currently reported. The numbers of children in each arm is a result and should not be included anywhere in the methods, e.g. in lines 121 and 123.

Response: Thank you for the comment. It has been incorporated in line no 141-156, page no.7-8 in clean version of the manuscript. The number of children in each arm has been deleted from the method.

For the analysis, first, a table of descriptive characteristics of the sample, without any statistical tests comparing the groups, should be presented. These should be means and standard deviations for continuous variables, and counts and proportions for categorical ones, in each group and overall. This should be Table 1. It should be followed by a table showing the mean and SE of continuous outcomes in each group and the difference in means, 95% confidence intervals and p-values, both crude and adjusted, from a linear regression model. For the binary outcome the table should have the counts of events and proportions in each arm and the odds ratio or risk ratio or risk difference with the 95% confidence intervals and p-values, both crude and adjusted. It is these results that should then be summarised in the text and abstract.

Response: Thank you for the suggestions. It has been incorporated in page no. 21-22 in clean version of the manuscript.

Please avoid using the ± designation anywhere in the text as it implies a range of values which is not what you seem to mean in each case where it is used.

Response: Thank you for the suggestion. It has been revised accordingly throughout the manuscript.

Please include the sources of funding and other support and the role of funders as a sub-section in the methods (e.g. the last subsection of methods).

Response: Thank you for the comment. It has been incorporated in line no 274-280, page no.13 in clean version of the manuscript.

Reviewer #3: 

This study is a good example for the effects of different kind of supplementation to ORS for children with acute infectious diarrhea. Randomization, patient selection and end-points are great for pediatric diarrhea study. However I cannot understand why the authors prefer to use this prebiotic for the treatment of diarrhea. Prebiotics have some beneficial effects on health (not innumerable), and majority of the effects of fiber are increased transit time (mainly proposed in children with constipation or other FGIDs). This may be beneficial for diarrhea if combined with probiotic strains.

Response: Thank you for the comment. As it is pointed out, the typical physiological function of dietary fiber is the gastrointestinal transit time. It has been reported that Fibersol-2, fermentable soluble dietary fiber, used in this study also has the effect of shortening the gastrointestinal transit time in pre-clinical experiments and human studies. When the functions of improving constipation and improving diarrhea could look like opposite, it may be open a question to examine the effect of Fiberol-2 on diarrhea. However, the intestinal environment is actually the important factor associated with both constipation and diarrhea. Up to today, it has been reported that ingestion of Fibersol-2 increases the numbers of beneficial bacteria such as bifidobacteria in the large intestine and increases production of short-chain fatty acids as metabolites. Those help to regulate good environment in the large intestine and Fibersol-2 is a prebiotic.

When acute infectious diarrhea is developed; the balance in intestine becomes temporarily impaired. To recover from diarrhea, it is effective to normalize the intestinal environment as soon as possible, and probiotics such as lactic acid bacteria are generally administered. On the other hand, soluble dietary fiber (hydrolyzed guar gum) has been reported to be effective in recovering from acute infectious diarrhea. The mechanism is considered that the hydrolyzed guar gum is a prebiotic that increases the numbers of bifidobacteria, regulates the intestinal bacteria, and increases beneficial metabolites such as short-chain fatty acids. 

Although Fibersol-2 is in the same category of prebiotic soluble dietary fiber as hydrolyzed guar gum, it has different physical properties and different fermentability by intestinal bacteria. Therefore, we have examined the effect of Fibersol-2 on acute infectious diarrhea for the first time in this study because Fibersol-2 has been reported to stimulate the growth of intestinal bacteria and affect the metabolites. 

We agree that there is a possibility to be more effective by the combination of prebiotics and probiotics. Since we investigated the effect of prebiotic Fibersol-2 alone this time, it would be interesting to see the effect of combination with probiotics in the future research.

---

## [Decision Letter · Decision Letter 1]

17 Jan 2022

PONE-D-21-01310R1Is Fibersol-2 efficacious in reducing duration of watery diarrhea and stool output in children 1-3 years old? A randomized, parallel, double-blinded, placebo- controlled, two arm clinical trialPLOS ONE

Dear Dr. Shahid,

Thank you for submitting your manuscript to PLOS ONE. After careful consideration, we feel that it has merit but does not fully meet PLOS ONE’s publication criteria as it currently stands. Therefore, we invite you to submit a revised version of the manuscript that addresses the points raised during the review process.

We look forward to receiving your revised manuscript.

Kind regards,

Ivan D. Florez, MD, MSc, PhD

Academic Editor

PLOS ONE

Journal Requirements:

Additional Editor Comments:

Reviewers have provided some comments that need to be addressed. Please prepare a revised version with a point-by-point response to their comments.

Reviewers' comments:

Reviewer's Responses to Questions

**Comments to the Author**

1. If the authors have adequately addressed your comments raised in a previous round of review and you feel that this manuscript is now acceptable for publication, you may indicate that here to bypass the “Comments to the Author” section, enter your conflict of interest statement in the “Confidential to Editor” section, and submit your "Accept" recommendation.

Reviewer #2: (No Response)

Reviewer #3: All comments have been addressed

2. Is the manuscript technically sound, and do the data support the conclusions?

Reviewer #2: Partly

Reviewer #3: Yes

3. Has the statistical analysis been performed appropriately and rigorously? 

Reviewer #2: No

Reviewer #3: Yes

4. Have the authors made all data underlying the findings in their manuscript fully available?

Reviewer #2: No

Reviewer #3: Yes

5. Is the manuscript presented in an intelligible fashion and written in standard English?

Reviewer #2: Yes

Reviewer #3: Yes

6. Review Comments to the Author

Reviewer #2: In the abstract, the lines reporting numerical results (41-42) should (1) for duration of watery diarrhoea and total watery stool output, instead of the statistics reported using ± which the authors don't indicate what they are, should report the difference in means with 95% confidence intervals in addition to the p-values; (2) for proportion recovered, the odds ratio (as stated in your statistical analysis section) with 95% confidence intervals in addition to the p-value. These can be obtained from appropriate regression models.

The statistical analysis is not appropriately described or presented. It should focus on (1) presenting the characteristics of participants in each treatment arm. Categorical variables should be presented as counts and proportions, and continuous ones as means with standard deviations or medians with IQRs as approporiate (this is usually reported in Table 1 and seems to have been done well). It is not normally necessary to conduct tests of normality of covariates; (2) the methods for estimating the treatment effects on the outcomes, with 95% confidence intervals and p-values. These would normally be appropriate regression models. The results tables coming out of these should report the count and proportion of binary outcomes in each arm, with unadjusted and adjusted effects, here odds ratios, with their 95% confidence intervals and p-values; for continuous outcomes, the means and standard errors (NOT standard deviations) of the outcome in each arm, along with the unadjusted and adjusted effect estimates (mean differences) with 95% confidence intervals and p-values; (3) how the sample size was determined - this is critial!

Thus, the results presented in Tables 2 to 4 are not appropriate. Please see https://www.sciencedirect.com/science/article/pii/S0140673618317823 for examples of typical approaches to presenting the results tables.

The statement in the statistical methods section about the power of the study is a result, and should be reported in the last paragraph of the results section.

Reviewer #3: (No Response)

7. PLOS authors have the option to publish the peer review history of their article (what does this mean?). If published, this will include your full peer review and any attached files.

Reviewer #2: No

Reviewer #3: No

---

## [Author Response · Author response to Decision Letter 1]

7 Jul 2022

Date: 7th July, 2022

To

Ivan D. Florez

From: 

Dr. Abu Sadat Mohammad Sayeem Bin Shahid

Corresponding Author

Subject: Response to the comments of the reviewer of PLOS ONE on manuscript Ref: PONE-D-21-01310-R2 titled “Is Fibersol-2 efficacious in reducing duration of watery diarrhea and stool output in children 1-3 years old? A randomized, parallel, double-blinded, placebo-controlled, two arm clinical trial.”

Dear Ivan D. Florez,

Thank you for evaluating our manuscript and providing us with the opportunity to submit the revised manuscript after addressing respected reviewer’s comments. We also express our sincere thanks to the respected reviewers for evaluating our manuscript. We are sending both the track change and clean versions of the manuscript that highlights the changes we have made from the previous version. We are also attaching this letter outlining a point-by-point response to each point kindly raised by the respected reviewer. 

We hope that our response will be appropriate to qualify the manuscript for publication in your well-reputed journal. 

We look forward to kindly hearing from you.

Thank you.

Funding statement

"This research study was funded by Matsutani Chemical Industry Company Limited, Japan on behalf of ADM/Matsutani LLC, USA. The funders didn’t have any role in study design, data collection and analysis, decision to publish, or preparation of the manuscript."

Responses to the comments of the respected reviewer

Reviewer #2: 

In the abstract, the lines reporting numerical results (41-42) should (1) for duration of watery diarrhoea and total watery stool output, instead of the statistics reported using ± which the authors don't indicate what they are, should report the difference in means with 95% confidence intervals in addition to the p-values; (2) for proportion recovered, the odds ratio (as stated in your statistical analysis section) with 95% confidence intervals in addition to the p-value. These can be obtained from appropriate regression models.

Response: Thank you for the comment. It has been revised accordingly (please see line no. 40-47, page no. 2-3 in track change version of the manuscript).

The statistical analysis is not appropriately described or presented. It should focus on (1) presenting the characteristics of participants in each treatment arm. Categorical variables should be presented as counts and proportions, and continuous ones as means with standard deviations or medians with IQRs as appropriate (this is usually reported in Table 1 and seems to have been done well). It is not normally necessary to conduct tests of normality of covariates; (2) the methods for estimating the treatment effects on the outcomes, with 95% confidence intervals and p-values. These would normally be appropriate regression models. The results tables coming out of these should report the count and proportion of binary outcomes in each arm, with unadjusted and adjusted effects, here odds ratios, with their 95% confidence intervals and p-values; for continuous outcomes, the means and standard errors (NOT standard deviations) of the outcome in each arm, along with the unadjusted and adjusted effect estimates (mean differences) with 95% confidence intervals and p-values; (3) how the sample size was determined - this is critial! Thus, the results presented in Tables 2 to 4 are not appropriate.

Response: Thank you for the valuable suggestions. It has been revised accordingly (Table 1-4 in track change version of the manuscript). Also see line no. 273-275, page no. 13 in track change version of the manuscript.

The statement in the statistical methods section about the power of the study is a result, and should be reported in the last paragraph of the results section.

Response: Thank you for the suggestion. It has been revised accordingly (please see line no. 337-339, page no. 15 in track change version of the manuscript).

---

## [Decision Letter · Decision Letter 2]

19 Oct 2022

PONE-D-21-01310R2Is Fibersol-2 efficacious in reducing duration of watery diarrhea and stool output in children 1-3 years old? A randomized, parallel, double-blinded, placebo- controlled, two arm clinical trialPLOS ONE

Dear Dr. Shahid,

Thank you for submitting your manuscript to PLOS ONE. After careful consideration, we feel that it has merit but does not fully meet PLOS ONE’s publication criteria as it currently stands. Therefore, we invite you to submit a revised version of the manuscript that addresses the points raised during the review process.

A number of outstanding concerns have been expressed regarding the statistical reporting in your study. Please respond carefully to these comments when preparing your revision.

We look forward to receiving your revised manuscript.

Kind regards,

Jamie Males

Editorial Office

PLOS ONE

Reviewers' comments:

Reviewer's Responses to Questions

**Comments to the Author**

1. If the authors have adequately addressed your comments raised in a previous round of review and you feel that this manuscript is now acceptable for publication, you may indicate that here to bypass the “Comments to the Author” section, enter your conflict of interest statement in the “Confidential to Editor” section, and submit your "Accept" recommendation.

Reviewer #2: (No Response)

Reviewer #3: All comments have been addressed

2. Is the manuscript technically sound, and do the data support the conclusions?

Reviewer #2: Partly

Reviewer #3: Yes

3. Has the statistical analysis been performed appropriately and rigorously? 

Reviewer #2: No

Reviewer #3: Yes

4. Have the authors made all data underlying the findings in their manuscript fully available?

Reviewer #2: No

Reviewer #3: Yes

5. Is the manuscript presented in an intelligible fashion and written in standard English?

Reviewer #2: Yes

Reviewer #3: Yes

6. Review Comments to the Author

Reviewer #2: There are still some problems with the statistical reporting in this manuscript. First, the sample size determination is repeated in two places: in lines 133 to 149 and again in lines 252 to 254.

That aside, the sample size determination is poorly reported. For example, when the authors say they anticipated a 25% relative reduction in 48-hour stool output, they do not indicate the anticipated stool output before or after the 48 hour period which would be useful in attempting to replicate their calculations; so it is not possible to do so. Furthermore, when they say that they assumed the proportion of children recovered within 72 hours would be 35% in the placebo group compared to 70% in the fibersol-2 group (or even the information about the duration of watery diarrhoea), there is no external information to justify why they thought these would be the case; it almost seems as if these numbers have been plucked out of thin air to justify the study size post-hoc.

The description of the statistical analysis is still quite poor. For example, the authors say in line 256: "in qualitative variables, differences in proportions were compared by the chi-squared test." I'm assuming this refers to categorical variables. First of all, the authors need to be clear whether these were the descriptive variables of the participants - in which case there would be no need for tests of significance - or outcome variables, in which case in my previous review I recommended performing regressions and reporting the magnitudes of association (odds ratios/risk ratios/risk differences - only one of these) with 95% confidence intervals and p-values, both unadjusted and adjusted where possible. This point also applies to the next statement in line 256 about 'quantitative data' (which I assume means continuous outcomes) - again here, if these were descriptive, then means and standard deviations are to be reported; if these are the outcomes, then means and standard errors in each group, followed by mean differences from linear regression models with 95% confidence intervals and p-values should be reported.

(adding this after reading the rest of the manuscript - some of the information I have asked for above is present in the manuscript, but the current description of the methods makes it seem like it has been done incorrectly)

Tables 1, 2, 3 and 4 are well-reported, and consistent with what one would expect - but not consistent with the statistical methods described, given my comments above. Table 5 is also useful additional information, but no mention of this in the statistical methods.

Reviewer #3: Recent version of the manuscript would be pıublish in PLos One, if Editorial Board also agree to publish.

7. PLOS authors have the option to publish the peer review history of their article (what does this mean?). If published, this will include your full peer review and any attached files.

Reviewer #2: No

Reviewer #3: No

---

## [Author Response · Author response to Decision Letter 2]

19 Dec 2022

Date: 19th December, 2022

To

Jamie Males

PLOS ONE 

From: 

Dr. Abu Sadat Mohammad Sayeem Bin Shahid

Corresponding Author

Subject: Response to the comments of the reviewer of PLOS ONE on manuscript Ref: PONE-D-21-01310-R2 titled “Is Fibersol-2 efficacious in reducing duration of watery diarrhea and stool output in children 1-3 years old? A randomized, parallel, double-blinded, placebo-controlled, two arm clinical trial.”

Dear Jamie Males,

Thank you for evaluating our manuscript and providing us with the opportunity to submit the revised manuscript after addressing respected reviewer’s comments. We also express our sincere thanks to the respected reviewers for evaluating our manuscript. We are sending both the track change and clean versions of the manuscript that highlights the changes we have made from the previous version. We are also attaching this letter outlining a point-by-point response to each point kindly raised by the respected reviewer. 

We hope that our response will be appropriate to qualify the manuscript for publication in your well-reputed journal. 

We look forward to kindly hearing from you.

Thank you.

Responses to the comments of the respected reviewer

Reviewer #2: 

Comment: There are still some problems with the statistical reporting in this manuscript. First, the sample size determination is repeated in two places: in lines 133 to 149 and again in lines 252 to 254.

Response: Thank you for the comment. We have now deleted the sample size determination statement from the statistical analysis section (please see line no. 248-250, page no.12 in track change version of the manuscript).

Comment: That aside, the sample size determination is poorly reported. For example, when the authors say they anticipated a 25% relative reduction in 48-hour stool output, they do not indicate the anticipated stool output before or after the 48-hour period which would be useful in attempting to replicate their calculations; so it is not possible to do so. Furthermore, when they say that they assumed the proportion of children recovered within 72 hours would be 35% in the placebo group compared to 70% in the fibersol-2 group (or even the information about the duration of watery diarrhoea), there is no external information to justify why they thought these would be the case; it almost seems as if these numbers have been plucked out of thin air to justify the study size post-hoc.

Response: Thank you for the valuable comment. We completely concur with you. Actually, sample size of 92 which was reflected in the manuscript was calculated on the basis of the potential stool output with proper reference. However, based on the previous recommendation from the respected reviewer, we also calculated the sample size based on other outcomes (duration of resolution of diarrhea and proportion of children recovered within 72 hours). Actually, we did not find any reference for these two outcomes and thus we calculated the sample size on the basis of assumptions. We want to admit that during the original design of the study ‘stool output’ was set as the lone primary outcome and other two were considered as secondary outcomes but in the final protocol somehow later two were also shown as primary outcomes. As we are not considering these two as primary outcomes, we have now revised the sample size determination statement (please see line no. 138-147, page no. 7 in track change version of the manuscript). 

Comment: The description of the statistical analysis is still quite poor. For example, the authors say in line 256: "in qualitative variables, differences in proportions were compared by the chi-squared test." I'm assuming this refers to categorical variables. First of all, the authors need to be clear whether these were the descriptive variables of the participants - in which case there would be no need for tests of significance - or outcome variables, in which case in my previous review I recommended performing regressions and reporting the magnitudes of association (odds ratios/risk ratios/risk differences - only one of these) with 95% confidence intervals and p-values, both unadjusted and adjusted where possible. This point also applies to the next statement in line 256 about 'quantitative data' (which I assume means continuous outcomes) - again here, if these were descriptive, then means and standard deviations are to be reported; if these are the outcomes, then means and standard errors in each group, followed by mean differences from linear regression models with 95% confidence intervals and p-values should be reported.

Response: Thank you for the valuable comments and suggestions. It has been revised accordingly (please see line no. 263-276, page no. 12-13 in track change version of the manuscript).

Comment: Tables 1, 2, 3 and 4 are well-reported, and consistent with what one would expect - but not consistent with the statistical methods described, given my comments above. Table 5 is also useful additional information, but no mention of this in the statistical methods.

Response: Thank you for the valuable comment. It has been revised accordingly (please see line no. 263-276, page no. 12-13 in track change version of the manuscript).

---

## [Decision Letter · Decision Letter 3]

12 Jan 2023

Is Fibersol-2 efficacious in reducing duration of watery diarrhea and stool output in children 1-3 years old? A randomized, parallel, double-blinded, placebo- controlled, two arm clinical trial

PONE-D-21-01310R3

Dear Dr. Shahid,

We’re pleased to inform you that your manuscript has been judged scientifically suitable for publication and will be formally accepted for publication once it meets all outstanding technical requirements.

Kind regards,

Samuel Bosomprah

Academic Editor

PLOS ONE

Additional Editor Comments (optional):

Reviewers' comments:

Reviewer's Responses to Questions

**Comments to the Author**

1. If the authors have adequately addressed your comments raised in a previous round of review and you feel that this manuscript is now acceptable for publication, you may indicate that here to bypass the “Comments to the Author” section, enter your conflict of interest statement in the “Confidential to Editor” section, and submit your "Accept" recommendation.

Reviewer #2: All comments have been addressed

2. Is the manuscript technically sound, and do the data support the conclusions?

Reviewer #2: (No Response)

3. Has the statistical analysis been performed appropriately and rigorously? 

Reviewer #2: (No Response)

4. Have the authors made all data underlying the findings in their manuscript fully available?

Reviewer #2: (No Response)

5. Is the manuscript presented in an intelligible fashion and written in standard English?

Reviewer #2: (No Response)

6. Review Comments to the Author

Reviewer #2: (No Response)

7. PLOS authors have the option to publish the peer review history of their article (what does this mean?). If published, this will include your full peer review and any attached files.

Reviewer #2: No

---

## [Editor Report · Acceptance letter]

20 Jan 2023

PONE-D-21-01310R3 

Is Fibersol-2 efficacious in reducing duration of watery diarrhea and stool output in children 1-3 years old? A randomized, parallel, double-blinded, placebo-controlled, two arm clinical trial 

Dear Dr. Shahid:

I'm pleased to inform you that your manuscript has been deemed suitable for publication in PLOS ONE. Congratulations! Your manuscript is now with our production department. 

Kind regards, 

on behalf of

Dr. Samuel Bosomprah 

Academic Editor

PLOS ONE